# Revitalizing from Ashes: Economic Development and Business Resilience in the City of Vukovar

Jakša Puljiz , Marina Funduk  and Ivana Biondić *

Institute for Development and International Relations, 10000 Zagreb, Croatia; jpuljiz@irmo.hr (J.P.); marina@irmo.hr (M.F.)
* Correspondence: ibiondic@irmo.hr

**Abstract:** The paper examines a paradigmatic example of post-conflict economic development of Vukovar, Croatia. It represents a pertinent case study for localities encountering analogous challenges, most notably urban areas in Ukraine in the near future. The war that broke out in 1991 brought significant human casualties, population displacement, and extensive destruction of residential, social, and economic infrastructure. The completion of the peaceful reintegration of Vukovar into Croatia's legal system in 1998 marked the beginning of the socio-economic revitalization process. The research scrutinizes the primary impediments and prospects for Vukovar's economic growth, probing why substantial investments in reconstructing housing, transport, communal infrastructure, and fiscal incentives for businesses have not paralleled its economic performance. It concentrates on the local business climate and influential factors as potential explanations for this discrepancy. The topic was designed as a case study and was covered by document analysis, survey method, and semi-structured interviews. Utilizing a mixed-methods approach, the study collates perspectives from entrepreneurs and business support institutions. The results confirmed that reconstruction of housing and social infrastructure is necessary, but more conditions are needed for successful post-conflict economic development, and that the business climate in lagging local units highly depends on state- and locally designed business-support measures.

**Keywords:** local economic development; post-conflict reconstruction; regional development; City of Vukovar; entrepreneurs

## 1. Introduction

Cities such as Vukovar are poignant symbols of resilience and the arduous path toward recovery in the aftermath of conflict. The City of Vukovar has undergone two simultaneous transitions in the post-1990 period. One refers to the typical transition of the political and economic systems for the ex-communist countries in Central and Eastern Europe. The other refers to post-conflict recovery after a war broke out in 1991 that destroyed the city's social and economic fabric. Such a "double transition" experience makes it an interesting case for Ukrainian cities seeking sustainable growth and prosperity once the war destructions are over. The case of Vukovar shows that return to prosperity can be challenging, depicting it as an ongoing process that continues to evolve even after three decades. From being praised as one of the most developed cities before 1991, Vukovar is today categorized as one of Croatia's less developed local units. Albeit slow, the city's progress is evident, as the study results will show. However, given the substantial amount of funding invested into post-conflict reconstruction, our research question arises: What factors impeded the City of Vukovar from achieving a more successful return to prosperity, and what were the primary constraints contributing to this outcome?

To answer this question, the paper examines the state of the economy in Vukovar and the factors influencing its post-conflict development. The study relies heavily on the position of entrepreneurs as key actors and pivotal contributors to local growth and

development (Lichtenstein and Lyons 2001; Audretsch and Keilbach 2005; Fritsch and Mueller 2008).

Consequently, our paper's main aim is to investigate the attitudes and opinions of entrepreneurs and entrepreneurial support institutions about the entrepreneurial environment and critical obstacles to more dynamic growth. It mainly considers the views of small and medium-sized entrepreneurs as the primary beneficiaries of various support measures. The study also questions the role of local authorities in promoting entrepreneurship in Vukovar. Previous analyses have affirmed that local authorities represent essential stakeholders in establishing a favorable business climate and supporting entrepreneurs (for example, Olsson et al. 2015; Olsson et al. 2020; Thekiso 2016). With that in mind, two hypotheses were identified and tested in this study:

**Hypothesis 1 (H1).** *Reconstruction of housing and social infrastructure is necessary, but more conditions are needed for successful post-conflict economic development.*

**Hypothesis 2 (H2).** *The business climate in lagging local units highly depends on state- and locally designed business-support measures.*

The structure of the paper unfolds as follows. It commences with a literature review and a short exploration of Croatia's main strategic acts and documents pertinent to developing the City of Vukovar. Following this, the methodology is presented, providing an account of the research methods employed. The paper's focal point lies in presenting the results derived from the online survey and semi-structured interviews, with the conclusive section dedicated to a thorough discussion of the analysis findings and the presentation of the key insights gleaned from the study.

## 2. Literature Review

### 2.1. Economic Transition and Regional Development in South-East Europe

To understand the current economic development of Vukovar, it is necessary to take a more comprehensive look at the transition process that ex-communist countries have undergone since 1990. The transition from a socialist system to capitalism in Eastern Europe was a complex and multifaceted process, characterized by a substantial initial decline in output, followed by deindustrialization and a rapid expansion of the service sector (Williamson 1993; Fisher and Sahay 2000; Lynn 2001). After several years, economic recovery ensued, buoyed by favorable geographic positioning, the availability of educated human resources, and substantial inflows of foreign direct investment (FDI). Hanzl-Weiss and Jovanović (2022) estimate that between 1993 and 2020, the average annual FDI inflow in Central, East, and South-East Europe constituted 4.4 percent of GDP, facilitating convergence with the EU average in terms of GDP per capita.

Despite sharing numerous commonalities, significant differences soon became evident among the former communist countries in Europe. Generally, countries in South-East Europe (SEE), including Croatia, lagged behind those in Central and Eastern Europe (CEE) in terms of the speed of economic recovery, structural change, and sectoral specialization during the first transition decade (Petrakos et al. 2002). The collapse of the socialist industrial base was more pronounced in SEE, mainly due to the dissolution of Yugoslavia and ensuing conflicts that impacted the entire region (Bartlett 2014). Moreover, SEE's privatization process was slower than that of CEE countries and the Baltic states. Persistent issues with corruption and weak institutions in SEE remained significant hurdles despite partial improvements attributable to the European integration process (Jurlin and Čučković 2010).

A salient aspect of the transition process was the exacerbation of regional disparities during the first transition decade (Petrakos and Totev 2008; Resmini 2002). Empirical studies indicate that post-1990 economic growth was spatially selective, favoring metropolitan and western regions, thereby intensifying pre-existing disparities (Fazekas 1996; Nemes-Nagy 2000; Bachtler et al. 2001; Romisch 2003; Petrakos et al. 2005). Regions heavily industrialized

during socialist times experienced the most severe declines, mainly due to the inability to restructure unproductive state-owned enterprises effectively in the early transition years (Bachtler et al. 2001; Ivanička and Ivanička 2007). Crisis management efforts related to the industrial restructuring of regions overly reliant on traditional industries were less successful than in more prosperous regions (Szalavetz 2003). This problem was closely linked to the significant social costs during rapid industry failures.

Croatia's political and economic transition occurred under turbulent circumstances following the dissolution of Yugoslavia and was compounded by the war environment that persisted until 1995 (Stojčić 2012). The disruption of transport links and loss of the Yugoslav market led to Croatia experiencing one of the most severe GDP declines among SEE countries in the early transition years (Vidovic and Gligorov 2006). By 1994, macroeconomic conditions had improved, primarily due to a successful anti-inflationary program and exchange rate stabilization, creating a more conducive environment for necessary economic reforms. However, the outcomes of these reforms, particularly in terms of the effects of the privatization process, were disappointing (Rohatinski and Vojnić 1996; Franičević 1999; Bendeković 2000). The decline in competitiveness of traditional industries in Croatia was prolonged, while the emergence of new industries was inadequate (Stojčić and Aralica 2018). The slow and inefficient industrial transformation was linked to deficiencies in the evolution of national industrial policy, which lacked elements of modern vertical industrial policy (Bartlett 2014). Post-conflict reconstruction posed an additional developmental and policy challenge, necessitating significant financial and other resources, particularly in the first decade after independence. Official estimates in 1998 prices put the total direct material and non-material war damages for Croatia at USD 37 billion, equivalent to 142 percent of Croatia's 1998 GDP. Indirect damages, such as production losses due to destroyed factories and facilities, were not accounted for, suggesting that total damages were substantially higher than estimated (State Commission 1999).

Croatia's regional development landscape is marked by the advantageous position of the City of Zagreb metropolitan region and some coastal counties where tourism is a critical economic activity. Conversely, lagging counties, predominantly in the Eastern region of Slavonia and Baranja, are characterized by a significant agricultural output share, a weak export base, lower education levels, and sustained outmigration (Đokić et al. 2016; Bačić and Aralica 2017). Regional competitiveness indicators for 2007 and 2010 identified this region as the least competitive, requiring concentrated long-term efforts to catch up with the rest of the country (Čučković et al. 2013). It is crucial to notice that this region was heavily impacted by the consequences of the Homeland War (1991–1995), with the City of Vukovar experiencing particularly severe destruction.

Even though more than thirty years have passed since the dissolution of Yugoslavia, specific studies on post-conflict economic recovery are rare. Bateman (2000) emphasizes the slow development of policies intended to promote small and medium enterprises in all ex-Yugoslav republics during the 1990s. He notes that "one-size-fits-all" neo-liberal policy interventions introduced in other post-communist economies were transferred to South-East Europe, slowing down the development of SMEs and the post-conflict recovery process. In their study of the role of the SME sector in the post-conflict economic recovery of Croatian counties, Maleković et al. (1998) advocate that support for small enterprises must be tied more closely to other development, reconstruction, and rehabilitation programs underway. They also see SMEs as an engine for local economic development and a tool to boost post-conflict and post-communist transition in lagging regions. Bartlett et al. (1996) point to the problem of the industrial base's collapse and the displaced's unwillingness to return as important obstacles for regions most affected by the conflict.

### 2.2. Pre- and Post-1991 Development Context of the City of Vukovar

The City of Vukovar, located in the easternmost part of Croatia along the Danube River, was classified as a middle-sized urban area in 1991, with a population of 46,735. It was known for its heavy industrialization, hosting Borovo, the largest shoe-making

company in Yugoslavia, and its rich cultural heritage, including a baroque-style city center (Karač 2004). The 1991 conflict brought immense human losses and physical devastation to Vukovar and its surrounding areas. The Vukovar-Srijem County, with Vukovar as its capital, accounted for 17.8 percent of total war damages, significantly exceeding its population share of 4.8 percent in 1991. Vukovar experienced material damage per capita, eleven times higher than the national average. The city's transport, utility, and energy infrastructure suffered extensive damage. Additionally, significant damage was inflicted on economic facilities in Vukovar, with Borovo and Vupik enduring the highest direct damages among all Croatian companies (Institute for Development and International Relations 2018).

From 1991 until the conclusion of the peaceful reintegration process in 1998, facilitated by the UN mandate (UNTAES mission), Vukovar remained outside of the control of the Croatian government. The 1998 reintegration marked the commencement of the reconstruction phase, with the city formally reintegrating into the Croatian legal system. Despite successful efforts in housing reconstruction and substantial investments in transport, utility, and energy infrastructure, the return of the displaced population saw only partial success. The return rate stood at 57.1 percent, falling below the county average of 71.1 percent (Živić 2012). Census data disclosed a notable decline in the population, reaching 30,126 in 2001 and continuing to decrease over the subsequent two decades, settling at 22,255 in 2021. This figure represents less than 50 percent of its pre-conflict size, underscoring the enduring demographic impact of the conflict.

In response, the central government designed a range of instruments to stimulate economic development, many of which remain in force. However, economic redevelopment proved to be an exceptionally challenging task. Economic growth was hampered by various challenges faced by local companies, such as market loss, outdated technology, employee excess, reduced property value, lack of credit support from commercial banks, unresolved ownership issues, slow privatization, delayed reconstruction of supporting infrastructure (especially in transport), a slow demining process, and the loss of a highly educated population (Faculty of Economics University of Split 2008). One of the most visible changes in Vukovar and other heavily industrialized Croatian cities is the employment structure, now dominated by small and medium enterprises (SMEs). The transformation is exemplified by Borovo, which, before 1991, employed around 23,000 employees, with approximately 16,000 in Vukovar and nearby municipalities. In contrast, in 2022, the same company employed only about 600 employees, according to data from the Financial Agency.

*2.3. Policy Instruments Related to Economic Redevelopment of Vukovar*

In 1996, the Croatian parliament passed two laws to foster post-conflict reconstruction, economic development, and return of the displaced people: the Act on Areas of Special State Concern (LASSC) and the Act on Reconstruction (LAR). LASSC contained many measures to encourage the return of local units' population, economic development, and fiscal capacity, which were therefore granted the status of Areas of Special State Concern. Most important were various tax concessions for businesses (profit tax and health and insurance contributions) and physical persons (income tax) (OG 44/199 (The Official Gazette 1996a)). LAR provided various measures for reconstructing and constructing damaged housing units (OG 24/1996 (The Official Gazette 1996b)). Both laws applied to Vukovar. In order to additionally stimulate and speed up post-conflict recovery in Vukovar, the Croatian Parliament passed the 2001 Act on Reconstruction and Development of the City of Vukovar, initially in 2001 (OG 44/01 (The Official Gazette 2001a)). Based on this Act, a new institution has been founded—Fund for Reconstruction and Development of the City of Vukovar (FRDV)—to provide additional funding for public and private development projects contributing to the city's economic and demographic revitalization. The fund's operations were based on the multiannual work program specifying the main obstacles for the city and the Fund's priorities. Investment areas covered by the Fund were initially quite diverse and included housing, communal infrastructure, transport, social infrastructure, cultural heritage, economic development, demining, human resources, spatial planning,

and cadastre. Several incentive measures were stipulated in other legal acts and programs, such as soft loans for entrepreneurs provided by the Croatian Bank for Development and Reconstruction (HBOR) or loan guarantees provided by the Croatian Agency for SMEs (HAMAG-BICRO).

According to the analysis of state measures for incentivizing the development of Vukovar prepared in 2008, between 1998 and 2007, over EUR 200 million were invested in the reconstruction of the housing stock (Faculty of Economics University of Split 2008). By 2014, the reconstruction of the housing stock had been nearly completed, covering over 95 percent of destroyed and damaged housing units (Fund for Reconstruction and Development of the City of Vukovar 2014). Along with the housing stock reconstruction, the central state invested heavily in the reconstruction of the physical infrastructure. Data from FRDV show that between 2009 and 2013, the value of realized investments into physical infrastructure was approximately EUR 23 million yearly. This was approximately four times the total value of the city budget at that time, indicating a very strong level of state support. Investments were mainly realized in social infrastructure (schools, museums, etc.) and local communal infrastructure (utilities, local roads, etc.), accounting for over 90 percent of total investments.

On the other hand, support for business development was relatively marginal, amounting to EUR 1.4 million annually (Fund for Reconstruction and Development of the City of Vukovar 2014). Equally important, the level of individual grant support for entrepreneurs was relatively modest, ranging between ten and twenty thousand euros. Support was mainly intended for the purchase of capital equipment and improvements of business facilities. A more recent report covering the period 2015–2022 shows that annual grant support for entrepreneurship has been reduced to EUR 480 thousand (Fund for Reconstruction and Development of the City of Vukovar 2022). On the other hand, the Fund's total budget has increased, meaning that the share of the Fund's support for entrepreneurship in its total activities has gradually declined.

Another wave of support that followed Croatia's accession to the EU was related to EU funding. For the period 2016–2023, Vukovar was entitled to EUR 20 million worth of investment in the program called the Intervention Plan. The plan was prepared under the guidance of the Ministry of Regional Development and EU funds and backed by the EU Cohesion policy funds. It envisaged support for implementing larger infrastructure projects, whose value was mainly over EUR 2 million and mainly in education, sport, and recreation (The City of Vukovar 2017). For the period 2021–2027, the city and five neighboring municipalities have prepared an Urban Development Strategy (The City of Vukovar 2023). The document provides a strategic framework for various investments, mainly into social infrastructure and transport, and opens the possibility of accessing EUR 18 million from the EU European Regional Development Fund to realize projects until 2030. This first strategic document directly promoted inter-municipal cooperation in strategic planning and project preparation.

To additionally support local entrepreneurs, from 2015 onwards, the city has prepared its yearly program for the development of entrepreneurship. The program usually encompasses a relatively high number of measures designed to support the various needs of entrepreneurs, from opening the business to increasing employment, purchasing equipment, developing skills, etc. Compared to the support provided by FRDV, this program has a smaller total budget and a smaller individual grant size, ranging mainly between three and six thousand euros (Institute for Development and International Relations 2019).

It is important to note that state support for Vukovar has been quite early based on the strategic documents referring to city development. From its establishment, FRDV based its activity on multiannual planning documents called Plan and Program for Reconstruction and Development of Vukovar. These documents were adopted for 2004–2008, 2009–2013, and 2014–2020. On the other hand, the City of Vukovar adopted its first development strategy only in 2010, while more effort in strategic planning by city authorities was realized after Croatia acceded to the EU. In entrepreneurship, the most important is the Economic

Development Strategy for the City of Vukovar 2021–2031 2022, which was adopted in 2021. The Strategy identified four strategic goals: (1) increasing the number of employees of entrepreneurs based in the City of Vukovar to 7.000; (2) increasing the average salary in entrepreneurship to 100 percent of the national average; (3) increasing the export value of micro, small and medium-sized enterprises by at least 150 percent compared to 2020; and (4) tripling of the annual value of entrepreneurs' investments in fixed assets compared to 2020 (Economic Development Strategy for the City of Vukovar 2021–2031 2022). However, unlike the Intervention plan and Urban development strategy, this document needs backing in specific and additional financial sources for the city. However, it relies on already secured financial means from national and local sources.

Table 1 summarizes the most important milestones in the case of support measures for the City of Vukovar.

**Table 1.** Key milestones in design of support measures for the City of Vukovar.

| Year | Main Acts and Strategic Documents | Coordinating Body | Main Support Measures |
|---|---|---|---|
| 1996 | Act on Reconstruction (OG 24/1996)—The Official Gazette (1996b) | Central state body in charge of reconstruction process | Housing reconstruction |
| 1996 | Act on Areas of Special State Concern (OG 44/1996)—The Official Gazette (1996a) | Central state body in charge of regional development | Other types of housing care for displaced population Tax reliefs for businesses Tax allowances for individuals |
| 2001 | Act on Reconstruction and Development of the City of Vukovar (OG 148/13)—The Official Gazette (2001b) | Central state body in charge of regional development/Fund for Reconstruction and Development of the City of Vukovar | Tax reliefs for businesses Tax allowances for individuals |
| 2002 | Plan and program for reconstruction and development of Vukovar (Fund for Reconstruction and Development of the City of Vukovar 2002) | Central state body in charge of regional development/Fund for Reconstruction and Development of the City of Vukovar | Investments into social and communal infrastructure (from large to small) Small grants for business development |
| 2017 | Intervention plan of the City of Vukovar (2016–2023) (The City of Vukovar 2017) | Ministry of Regional Development and EU funds/City of Vukovar | Large investments into educational, sport, and communal infrastructure. Grants for business development |
| 2015 | City's program for support of entrepreneurship (on yearly basis) | City of Vukovar | Small grants for entrepreneurs |
| 2023 | Urban development strategy 2021-2027 (The City of Vukovar 2023) | Ministry of Regional Development and EU funds/City of Vukovar | Large investments into cultural heritage, urban mobility, city transport, sport and social infrastructure |

Source: Authors' elaboration.

## 3. Methodology

The research topic was designed as a case study, aligning with Yin's (2014) methodology. The primary methods for gathering and processing information included document analysis, survey, and semi-structured interviews.

Data on the City of Vukovar was collected after examining the relevant academic and policy literature. Secondary data was mainly collected from the Croatian Bureau of Statistics, the Ministry of Regional Development and EU funds, and the Financial Agency (which collects yearly financial reports on Croatia's business entities). Then, to gain further insights into the state of entrepreneurship in the city, the opinions and attitudes of entrepreneurs were examined through an online survey. A structured questionnaire was

prepared containing questions regarding the business climate in Vukovar, the collaboration between entrepreneurs and the local government, key issues, and essential economic needs. One hundred two entrepreneurs completed the survey and responded to the questionnaire. Informed consent was obtained from all participants, and their identities remained anonymous throughout the study.

A mixed-methods approach was used to analyze the data from the survey. Several considerations guided the decision to use a mixed-methods approach and integrate quantitative and qualitative data. The survey responses of entrepreneurs supplemented quantitative data from the business registers and the Croatian Bureau of Statistics, while semi-structured interviews with entrepreneurs and other local stakeholders provided qualitative data. The use of multiple methods allowed for validation through data triangulation. Findings from one method could be cross-verified or complemented by findings from another, enhancing the robustness of the study's outcomes. Quantitative data added a layer of precision to the research, enhancing the ability to draw meaningful insights and contribute to a more thorough comprehension of the subject matter. In contrast, qualitative data were vital for confirming and explaining these patterns by examining individual responses and narratives. Moreover, qualitative data, particularly from open-ended questionnaire responses, provided an additional contextual layer to interpreting quantitative results.

Individual interviews were conducted through in-depth discussions with twenty entrepreneurs operating in Vukovar in November 2021. Interviewees were selected based on the criteria that their headquarters is registered in the City of Vukovar, that they have operated longer than five years, and that they are sectorally balanced (most were from manufacturing, followed by various service sectors). The interviews were mainly achieved through face-to-face meetings, while two were conducted via phone conversations due to the inability of selected entrepreneurs to attend in person. Entrepreneurs had the opportunity to provide comments and opinions through a structured conversation based on prepared and pre-distributed questions, and a high-quality discussion unfolded beyond the formal scope of the questions. Interviews were also conducted with representatives of entrepreneurial associations: the Croatian Chamber of Economy—Vukovar Branch and the Croatian Chamber of Crafts—Craftsmen Association of the City of Vukovar. Four extra interviews were conducted with representatives of the Vukovar Development Agency—VURA, the Fund for the Reconstruction and Development of the City of Vukovar, the University of Applied Sciences "Lavoslav Ružička" in Vukovar (Polytechnic Vukovar), and the Branch Office of the Croatian Employment Service in Vukovar throughout December 2021 and January 2022. All respondents from business support institutions and education had a long work history and extensive knowledge of the city. In total, 26 interviews were conducted with stakeholders involved in the local labor market.

## 4. Results

### 4.1. State of Local Socio-Economic Development

As previously explained, the conflict significantly impacted the local economy in Vukovar. Since the end of reintegration in 1998, economic activities have also started to recover. However, reaching the pre-conflict level in some aspects, such as employment level, turned out to be non-achievable, directly related to the unsuccessful return process of the displaced population and substantial outmigration in the post-conflict period.

Figure 1 illustrates the evolution of the employed population relative to the total population in Croatia and Vukovar from 1988 to 2021. By the year 2021, the employment-to-population 20–59 years ratio in Croatia had not only recuperated but exceeded its pre-conflict levels. In contrast, the City of Vukovar exhibited a markedly different trend, with its ratio only slightly surpassing pre-conflict figures. Examination of the absolute employment figures underscores the extent of the devastation the workplaces experienced. In Vukovar, the employee count plummeted from over 30,000 in 1988 to less than 10,000 in 2015, representing approximately a two-thirds decline. Since 2016, employment figures have steadily increased to around 12,600 in 2021 but are still significantly lower than in 1988.

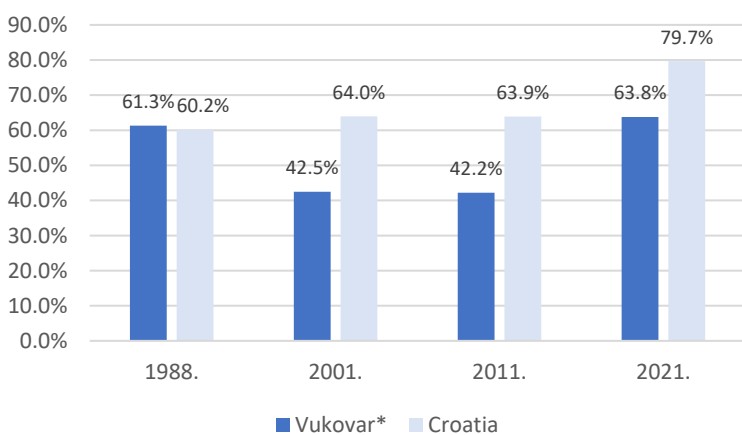

**Figure 1.** Ratio of employed people relative to population 20–59 years. Source: Authors' calculation based on data from Croatian Bureau for Statistics (Croatian Bureau of Statistics Database n.d.). For 1988, only employment data were collected, while data on population 20–59 were based on population census figures from 1991. * Estimates for Vukovar are made based on the territorial borders of the city from 1988 in order to ensure sound comparison between 1988 and other years.

The employment record demonstrates that post-conflict economic recovery in the case of Vukovar is a highly challenging process and that the enormous investments into reconstruction have not led to desired economic outcomes, at least in terms of employment.

One of the explanations for the unsatisfactory employment level, even twenty years since the beginning of the reconstruction process, is a lack of investors, especially foreign ones. Derado et al. (2011) confirm a positive relationship between the economic performance of Croatia's regions and the amount of inward foreign direct investments (FDI). However, while some other localities in Croatia have benefited enormously from the foreign direct investments, Vukovar and its county were relatively unsuccessful in the period 2002–2011 (Kersan-Škabić and Tijanić 2014). Data acquired from the Financial Agency on entrepreneurs employing more than 20 workers for 2022 corroborate previous findings, as they show that out of forty-three companies, only three were foreign-owned, confirming the small role of FDI for the city's economy, despite the incentives and favorable geographical position along the Danube and between Croatian capital of Zagreb and Serbian capital of Belgrade (Financial Agency Database 2023). This indicates that Vukovar exhibits persistent issues with attracting foreign investors despite numerous incentives provided.

The performance of the business sector has varied over time. Results from Table 1 demonstrate that business entities have been relatively successful in the first period until the economic crisis in 2008 broke out. Entrepreneurs have significantly increased their revenues, gross investments, and number of employees, achieving above-average results compared to the national level. However, they could not achieve above-average wage growth, an essential aspect for overall development, given that the average wage is continuously below average. According to data for 2020, the average wage in the business sector was still around 20 percent less than the national average. The 2008–2015 period was challenging due to the prolonged economic crisis that hit Croatia. In such circumstances, business entities from Vukovar recorded a substantial decline in investments and employment, indicating a lack of resilience to deal with the economic crisis. Strong revenue growth has been related to exceptional positive dynamics of one trading company operating within the energy sector. However, its influence on local employment remains relatively minor. The post-crisis period from 2015 onwards brought the city back to above-average dynamics for all considered indicators. Particularly important, wages have recorded the most robust growth so far, reducing the wage level gap compared to the national average.

Improved business performance in the post-2015 period is correlated with the improved perspective of living in Vukovar. As Figure 2 shows, Vukovar experienced a negative migration trend for most of the post-conflict period, with people migrating to

other Croatian counties and abroad. However, the situation changed over time, especially since 2017, closely resonating with business dynamics, albeit with a time lag, as can be noticed when comparing Table 2 and Figure 2. After the period of extensive negative net migration, the migration balance abroad was practically neutral in 2022.

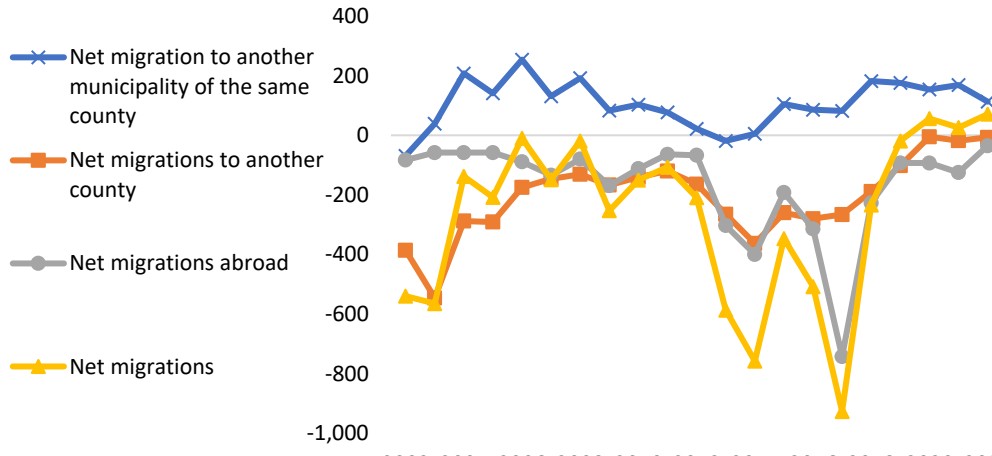

**Figure 2.** Internal and external migration balance for the City of Vukovar, 2001–2022. Source: Authors' calculation based on data from Croatian Bureau for Statistics (Croatian Bureau of Statistics Database n.d.).

**Table 2.** Performance indicators of business entities from the City of Vukovar, 2003–2020.

| Indicator | Dynamics (Absolute Values) | | | Dynamics in Comparison to National Average (Croatia = 100) | | |
|---|---|---|---|---|---|---|
| | 2003–2008 | 2008–2015 | 2015–2020 | 2003–2008 | 2008–2015 | 2015–2020 |
| Number of enterprises | 45.9% | 35.5% | 40.9% | 118.8 | 106.2 | 108.0 |
| Number of employees | 35.7% | −31.7% | 36.8% | 120.4 | 72.9 | 121.1 |
| Revenues | 100.8% | 131.9% | 113.8% | 138.3 | 237.7 | 183.8 |
| Average wage | 13.1% | 22.7% | 29.1% | 92.6 | 103.3 | 108.5 |
| Gross investments into long-term assets | 242.1% | −77.9% | 0.6% | 222.8 | 43.4 | 151.4 |

Source: Authors' calculation based on data from Financial Agency.

Besides continuous improvement in terms of the quality of social infrastructure, enhanced availability of better-paid jobs is another potential explanation for such migration dynamics. The relationship between incomes and migration is thus further explored by comparing dynamics in net personal incomes per capita (relative to the national average) and net migrations (absolute values). In this case, personal incomes include workplaces in the business and non-business sectors. Results from Figure 3 show that there is indeed a familiar pattern, albeit with a time lag in case of migrations. For example, the relative fall in personal incomes in 2010–2012 accompanied the fall in migration in 2011–2014. Similar lagged reactions in migration data are observable between 2016 and 2019 when relative growth in personal incomes took place. However, the positive reactions on the migration side were evident from 2017 onwards.

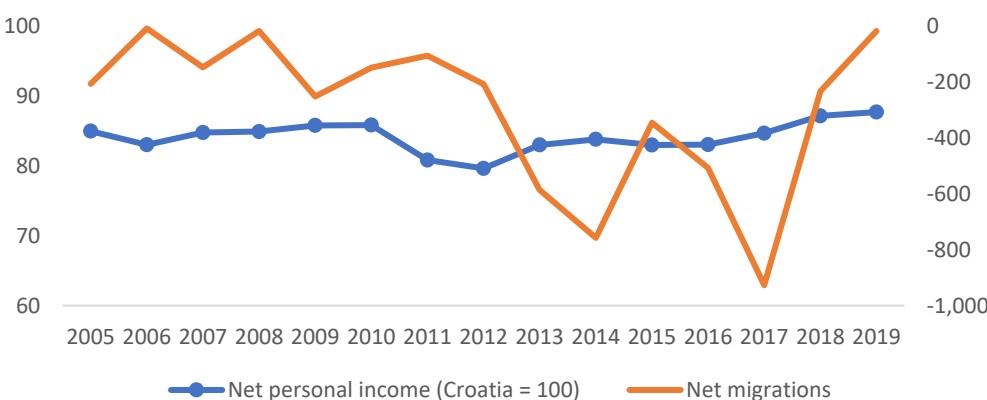

**Figure 3.** Net migrations and personal incomes for the City of Vukovar, 2005–2018. Source: Authors' calculation based on data from Ministry of Regional Development and EU funds (personal incomes) and Croatian Bureau of Statistics (migrations) (Croatian Bureau of Statistics Database n.d.).

Except for secondary data analysis, surveys and interviews with relevant stakeholders were conducted to complement initial findings further and deepen the comprehension of the business environment in Vukovar.

*4.2. Survey Findings*

The analysis of responses from entrepreneurs from the online survey provides further insights into their perspectives on the entrepreneurial environment in Vukovar and key obstacles to growth. The online survey, whose questions can be found as Appendix A, was sent to 593 entrepreneurs from the City of Vukovar, with a response rate of 17.2% (102 filled out the forms received). Overall, the prevailing sentiment among entrepreneurs on the business environment is moderately positive, with an overall score of 3.22 out of 5 (where one marks a lousy business environment and five marks an excellent business environment).

Several vital challenges faced by entrepreneurs were identified from the survey. The shortage of a qualified workforce is a pervasive issue, limiting the potential for further development and expansion. Coupled with this is the challenge of low local purchasing power, intricately linked to demand dynamics and constraining the regional entrepreneurial ecosystem. High operating costs, incredibly impactful for lower-value-added production reliant on utility prices, compound the challenges businesses face in Vukovar. The results can be seen in Figure 4.

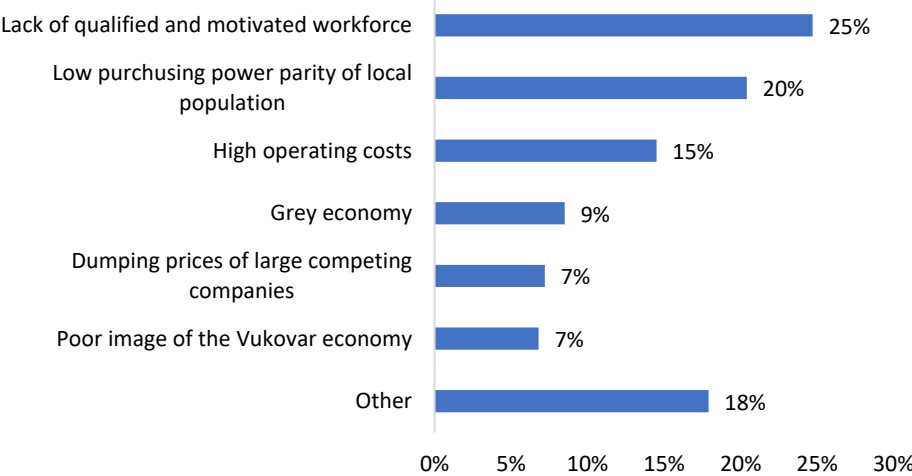

**Figure 4.** Most important challenges faced by entrepreneurs in business (% of total votes). Source: Authors' calculation based on data from the survey (N = 102).

The fall of the importance of the manufacturing industry, but also the lack of belief in its future, is acknowledged when looking at the entrepreneurs' view on the most promising sectors (Figure 5). IT, tourism, food, and agriculture sectors were recognized as leading sectors for future growth. The IT sector, in particular, stands out as a beacon for potential economic development, buoyed by favorable conditions like a 50 percent refund of payroll taxes. However, it is acknowledged that to realize this potential fully, there is a need for a qualified and available workforce, high-quality broadband internet coverage, and targeted campaigns to position Vukovar as an attractive destination for IT businesses. Tourism is identified as another sector with high growth potential, calling for a shift from current "daily tourism" to a more extended stay model. Entrepreneurs see an opportunity for expansion by enhancing offerings in hospitality and gastronomy, branding the destination, and organizing seasonal programs and events. The agricultural-food sector also emerges as a sector with high expectations for future growth, underlining the diverse opportunities present in Vukovar.

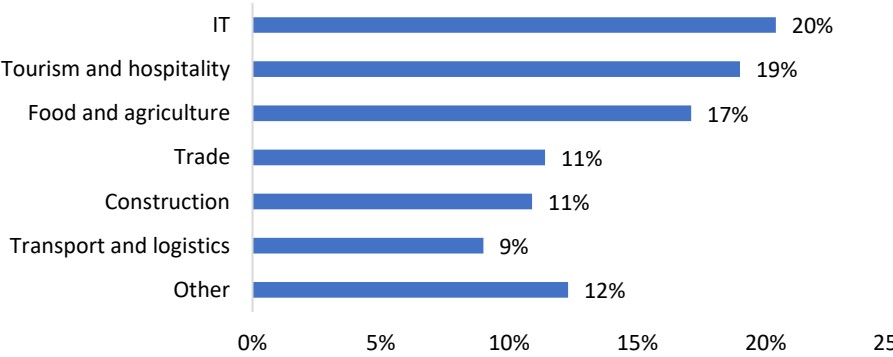

**Figure 5.** Sectors—carriers of the economic development of Vukovar in the next decade (% of total votes). Source: Authors' calculation based on data from the survey (N = 102).

The survey also explores the most crucial local factors for enhancing entrepreneurship in Vukovar, as shown in Figure 6. Fiscal incentives outlined in the Act on the Reconstruction and Development of the City of Vukovar take precedence, indicating the significance of these policy instruments for local entrepreneurs. The quality of city support programs and those from the Fund for the Reconstruction and Development of the City of Vukovar are also highlighted, as well as the possibilities for using EU funds. On the other hand, the quality of the local utility and educational infrastructure, as well as the proactive attitude of local government, were given much lower scores, suggesting that direct fiscal incentives are still perceived as the best stimulus for entrepreneurs. Such a view of the entrepreneurs confirms their strong attachment to the subsidies, which is hardly surprising given that some have been present for over two decades.

The overall rating for the cooperation between the City of Vukovar and entrepreneurs is relatively high (average grade 3.7). While many interviewees express high satisfaction levels, improvement is needed, particularly in communication and networking support. Entrepreneurs also point to the need for coherence and complementarity between public calls for entrepreneurs prepared by the FRDV and those prepared by the City of Vukovar. This made it challenging for entrepreneurs to determine precisely what to apply for and which parts of their activities would be eligible.

The survey also revealed that despite the prolonged presence of the local entrepreneurship program as well as a range of other activities by the city and local entrepreneurship agency, almost half of the respondents rarely or never cooperate with the city, underscoring the need for enhanced communication and visibility of the city's activities. Those who were participating in the annual public call for entrepreneurship development (published for 2021) were, on average, moderately satisfied with its main features and level of administrative complexity (the average grade for the program's main features varies between 3.18

and 3.38), meaning that the program itself can be further improved to better suit the needs of participants.

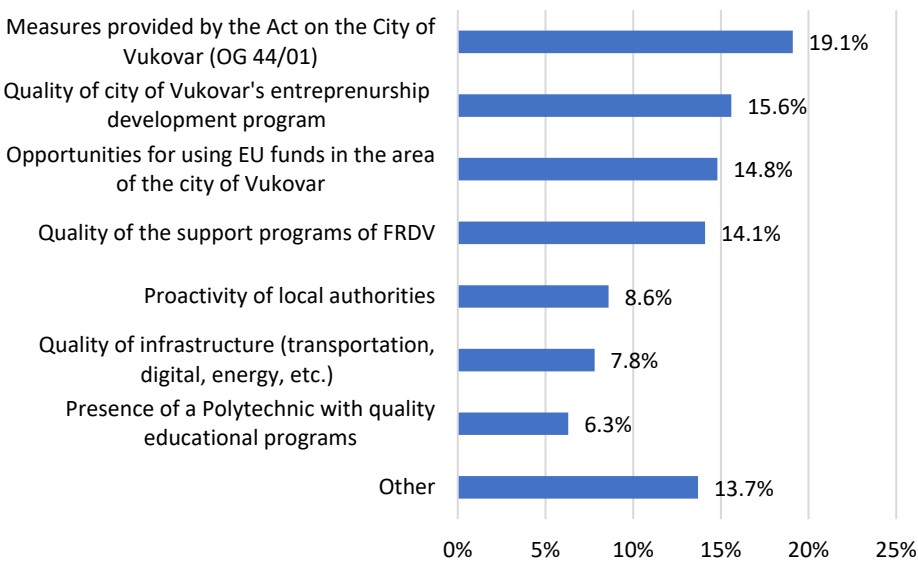

**Figure 6.** Most important local factors for further development of entrepreneurship in Vukovar (% of total votes). Source: Authors' calculation based on data from the survey (N = 102).

*4.3. Interview Findings*

Semi-structured interviews with key stakeholders, including entrepreneurs and entrepreneurial organizations, were organized once the survey had been carried out. The question guidelines can be found in Appendix B.

The findings regarding the business environment and conditions in Vukovar generally align with the results obtained from the online survey. Most interviewees expressed satisfaction with profit tax exemptions, concessions for health and pension contributions, and measures from the programs provided by FRDV and the City of Vukovar. A smaller part of them perceived the business environment as challenging, citing meager wages and continuous emigration as signs of weaknesses. Also, several interviewees emphasized the burden of the post-war city with a need for a positive city image outside of the war-related topics. Lack of greenfield investments is widely recognized as an essential issue. Several interviewees connect this problem, among others, with the need for more information about the tax and other benefits for potential investors in Vukovar (those coming from other regions or from abroad). Another frequently mentioned issue was the need for more networking among local entrepreneurs. There are no regular business events that would help share experiences and matchmaking, which could be valuable, especially for young entrepreneurs. Also, the city does not host any significant business events that would be of national or international significance and could be used for the city's promotion as a good site for business. Such commentaries confirm that more indirect and non-financial support activities are also an important aspect of promoting the local business climate and that local government should pay more attention to it.

Concerning pressing obstacles for business, respondents highlighted issues like a lack of business premises and storage spaces in the city, a chronic shortage of qualified and adequate workforce, low purchasing power, and a relatively narrow local market. Many find the problem of the quality and coverage of educational programs offered in the city, emphasizing the need for IT-related education and the low quality of education in general. They see the presence of Polytechnic Vukovar as the only higher educational institution with local presence as an essential factor for the city's development. However, they do not find sufficient benefits for their businesses given the content of the educational programs provided by the Polytechnic.

The Croatian Employment Service representative noted that many unemployed individuals need more interest in finding a job and decline available opportunities, partially related to the lack of entrepreneurial spirit. Quality of infrastructure, including internet connection and transport infrastructure, received positive ratings overall. However, concerns were raised about transportation links with neighboring rural areas, with bus lines needing to meet those areas' needs adequately. This creates a problem for daily migrants from rural areas, especially women, who often use public transportation to reach the city.

Cooperation with local administration yielded mixed comments, with some expressing contentment. In contrast, others emphasized dissatisfaction, expecting more support from the city and better communication of the city with the business community. However, existing incentive measures were consistently highlighted as valuable instruments driving entrepreneurship in Vukovar. Several participants expressed that local administration should be more active in targeting and attracting potential investors, given the benefits they can enjoy in Vukovar.

The perspective on entrepreneurship in Vukovar and its development leaned predominantly positive, with many entrepreneurs identifying EU funds as crucial support sources for the upcoming period. However, most entrepreneurs emphasized that plans for further growth and expansion were contingent on skilled staff availability and suitable business premises.

## 5. Discussion

Research findings offer an insight into the post-conflict entrepreneurial landscape for a small urban area represented by the City of Vukovar. After its reintegration into Croatia's legal system, the city has experienced enormous public investments in its reconstruction. These investments mainly focused on rebuilding housing stock, transport, and social and communal infrastructure. However, it became evident that these investments did not yield the expected economic outcomes even twenty-five years after the reconstruction started. From one of the most developed cities before 1991, Vukovar is among the less developed Croatian cities today. The new economic structure built after 1998 could not provide sufficient well-paid jobs. Consequently, new waves of emigration took place, the strength of which largely depended on the national economic cycle.

Policy instruments to spur entrepreneurship were manifold and included nationally and locally designed ones. The approach gradually evolved from the initial emphasis on tax concessions to become more program-based, providing grants for investments and other needs of the entrepreneurs (Fund for Reconstruction and Development of the City of Vukovar in 2004 and the City of Vukovar only in 2015). Today, it involves a multiplicity of different support measures (mainly provided by the city's program for entrepreneurship) covering various needs of entrepreneurs. While the increasing availability of various support measures is a positive indicator of an improved business climate, questions arise regarding the effectiveness of the incentive programs in achieving long-term sustainability, fostering innovation, and addressing the specific needs of different sectors. Most of the grants awarded to entrepreneurs by the FRDV and the City of Vukovar were relatively small and tailored mainly to the needs of start-ups and micro enterprises (Institute for Development and International Relations 2019). Such a fragmented approach to grant incentives seems inefficient in stimulating larger capital investments with a more significant impact on the local economy. Data on foreign investments confirm that attracting investors from abroad has not been successful, putting additional doubts on the effectiveness of incentives in attracting investors.

Survey shows stakeholders see more perspective for future development in less-capital-intensive sectors such as IT. However, this contradicts the availability of educational programs at the local level and the developments in other nearby cities. The only higher education institution in the city does not provide such education (University of Applied Sciences "Lavoslav Ružička" in Vukovar), so the closest higher education related to Information and communications technology (ICT) is in the nearby regional and university center,

the City of Osijek. Osijek has been a well-established regional ICT center for extended periods, intensively working on creating its own ICT entrepreneurial ecosystem (Mesarić et al. 2014). In such circumstances, relying on the ICT sector as a driving force seems to lack foundations.

While the efforts of the city administration are generally well-received among entrepreneurs, there is plenty of room for improvement. Enhanced coordination and clear communication channels between entrepreneurs and local authorities could foster a more dynamic and responsive entrepreneurial ecosystem. It is also evident that the city's benefits for potential investors need to be better promoted. Neither general nor targeted campaigns towards potential investors have been undertaken so far, consequently reducing the true potential of fiscal and other benefits in attracting investments. This is surprising given the vast amount of evidence that investment promotion measures can bring tangible benefits (Charlton and Davis 2007; Drahokoupil 2008; Pasquinelli and Vuignier 2020) and the important role of the subnational level in investment promotion (OECD 2018). Another important issue is the need for adequate business premises, which can deter the growth of local entrepreneurs and slow the arrival of new investors. This issue is closely linked with land management and the city's spatial planning practice, which seems to represent a policy challenge for the local administration.

Responding to our research Hypothesis 1 (H1), reconstruction of housing and social infrastructure are necessary but insufficient conditions for successful post-conflict economic development; our case study reveals that despite a very successful reconstruction effort and initial partial return of displaced population, Vukovar very quickly started to face substantial outmigration and loss of population. This indicates that the economic dynamism achieved in the first phase of economic redevelopment (until 2008) was too weak to secure favorable conditions. The reasons for such an outcome should be sought, among others, in the influence of policy instruments since they were the main impetus in rebuilding the local economic fabric. Emphasis on tax and other concessions was obviously not attractive enough to stimulate investors to open more and better-paid workplaces. Program-based grant support for entrepreneurship came later and was mainly focused on smaller grants, thus reducing its potential to attract more significant investments. The survey also affirms that intensive support from national and local levels did not solve some pressing issues, such as lack of qualified workforce or lack of business premises. This raises additional questions about the appropriate design of support instruments.

As a result, the city today is still lagging behind most of its peers in Croatia, as seen by recent results on the development ranking of local self-government units. According to results published in January 2024 published by the Ministry of Regional Development and EU funds, Vukovar was ranked 260th place among 556 local units in Croatia, distinctively lower than most of the local units of similar population size and far from its pre-conflict status and far away from its pre-1991 status as one of the most developed cities in Croatia (Ministry of Regional Development and EU Funds 2024). As such, the results support Hypothesis 1 (H1) of our research.

Research findings also support Hypothesis 2 (H2), which asserts that business climate in lagging local units highly depends on the combination of state and locally designed business-support measures. Survey and interview results confirmed that entrepreneurs find support measures essential for their business despite most measures having a limited financial impact. Given the constrained fiscal capacity of local authorities in the post-conflict period, it is clear that central state measures are of pivotal importance, particularly in the beginning, until the city becomes more fiscally capacitated. While recognizing the importance of direct fiscal support, entrepreneurs expressed the necessity for more support in networking and the general promotion of entrepreneurship through various events and other activities. Such soft activities could help improve the city's general image, which is still best known for its destructions during the war. The limited emphasis on robust networking platforms poses challenges for local entrepreneurs, hindering knowledge sharing and collaboration. To fortify Vukovar's entrepreneurial ecosystem, urgent attention

is needed to formulate and implement measures enhancing networking opportunities, improving communication channels, and promoting cross-sectoral exchange, fostering a more dynamic and interconnected business environment that attracts external interest and investments for sustained economic growth. The recommendations by entrepreneurs and other stakeholders engaged in entrepreneurship should serve as a guiding tool for both local and national actors in formulating the next set of support measures. Central state support measures have remained relatively static, maintaining practically the same content since their introduction. On the other hand, the City of Vukovar, through its program, demonstrated a more innovative approach to designing measures but faced limitations in financial support, hindering its overall effectiveness.

## 6. Conclusions

Numerous local units in Croatia have passed through difficult transitions in the last three decades, combining the post-war reconstruction with the need to transform their local economic structure and improve their long-term resilience to contemporary economic and social challenges. The City of Vukovar is a distinguished example of such development, given the level of its pre-war status and the extensive war-related destructions it has encountered. Its prolonged transition is not only connected to the consequences of war but also to its heavy industrialization before the war, which is in line with the findings from earlier studies (Bachtler et al. 2001; Ivanička and Ivanička 2007; Bartlett et al. 1996). Along with creating necessary conditions for the return of the displaced populations in terms of housing and communal infrastructure, post-war economic recovery has to be addressed with equal policy importance from the very beginning, as has been suggested by Maleković et al. (1998). Vukovar's experience demonstrates that tax and other concessions are insufficient in creating a favorable local business climate and that more creative policy solutions are required.

Close cooperation between central and local levels is particularly important given the lack of necessary financial and human resources in war-affected areas. Croatian case of setting up a special centrally managed institution to spur Vukovar's socio-economic development turned out to be a step in the right direction. However, to ensure effective contribution from such institutional support, it is necessary to continuously adapt its instruments to the evolving needs of the local economy. In that sense and in line with conclusions by Kwon and Gonzalez-Gorman (2014), the study suggests the importance of continuously evaluating entrepreneurs' evolving needs and the adaptive refinement of program content to align with emerging requirements. This recommendation underscores the dynamic nature of the entrepreneurial landscape, emphasizing the necessity for ongoing support systems to evolve in tandem with the changing needs of businesses. By regularly assessing entrepreneurs' challenges and tailoring support programs accordingly, stakeholders can ensure the sustained relevance and effectiveness of initiatives that foster entrepreneurial growth in Vukovar. This proactive approach not only enhances the resilience of the local business ecosystem but also contributes to creating a supportive environment conducive to innovation and sustainable development.

As recent research by Rodzinka et al. (2023) shows, in the case of Polish subnational units, there is no single instrument that proved successful in spurring local economic development. Instead, different sets of instruments to support entrepreneurship should be chosen for different places depending on their structural characteristics.

While this research effectively provides valuable insights into entrepreneurs' perspectives on the entrepreneurial environment and identifies key obstacles to growth, it is crucial to acknowledge and address certain limitations inherent in the study. The study's scope is confined to Vukovar, a mid-size city in Croatia with a population of 23 thousand people as per the 2021 census. While the insights gained from this setting are valuable in their own right, caution must be exercised when extrapolating these findings to larger cities in different geographical contexts, such as Ukraine. Notably, Vukovar's size may need to adequately mirror the scale and complexity of more substantial Ukrainian cities, where

a city of comparable size might be considered small. Also, a more extensive and diverse pool of interviewees would additionally add to a better understanding of entrepreneurial challenges and opportunities.

Despite these limitations, this research gives ground for further exploration of the topic and highlights areas where research could provide additional depth and context. Future research could delve more deeply into exploring demographic challenges and the interplay between demographic factors and local economic development. Investigating more in-depth reasons for the working-age population to settle or leave the city can bring further insights that contribute to formulating sustainable strategies for fostering economic resilience and vitality in Vukovar. This demographic lens can shed light on the evolving composition of the local populace and its implications for entrepreneurship, employment, and community well-being.

**Author Contributions:** Conceptualization, M.F. and I.B.; methodology, I.B. and J.P.; validation, M.F.; data curation, I.B. and J.P.; writing—original draft preparation, M.F., I.B., and J.P.; writing—review and editing, M.F., I.B., and J.P.; visualization, I.B.; supervision, M.F.; project administration, J.P. and I.B.; funding acquisition, J.P. All authors have read and agreed to the published version of the manuscript.

**Funding:** This research was funded by the City of Vukovar, a contract for the procurement of services for the preparation of the Economic Development Strategy of the City of Vukovar; class: 300-01/21-01/1, registry number: 02-100/031-21.

**Institutional Review Board Statement:** Ethical review and approval were waived for this study, since the research was not medical research on human subjects and did not include identifiablehuman material and data. It collected research participants' opinions and attitudes.

**Informed Consent Statement:** Informed consent was obtained from all subjects involved in the study.

**Data Availability Statement:** Data are contained within the article.

**Conflicts of Interest:** The authors declare no conflict of interest. The funders had no role in the design of the study; in the collection, analyses, or interpretation of data; in the writing of the manuscript, or in the decision to publish the results.

### Appendix A. Questions from the Online Survey

1. Year of establishment of the company: _______
2. Please indicate the category to which your company belongs based on the number of employees.

    - Micro (<10 employees)
    - Small (<50 employees)
    - Medium (<250 employees)
    - Large (>250 employees)

3. How often do you collaborate with the City of Vukovar?

    - Annually
    - Every two to three years
    - Rarely (once every four years or less)
    - Never

4. How would you generally rate the quality of your collaboration with the City of Vukovar (1–5, where 1—Poor collaboration, 5—Excellent collaboration)
5. Please explain your previous answer. ___________________________________
6. How would you rate the business environment in the City of Vukovar (1–5, where 1—Poor business environment, 5—Excellent business environment)
7. Please select up to three sectors that you believe will be drivers of economic development in Vukovar in the next decade:

    - IT sector
    - Agricultural and food sector

- Transportation and logistics sector
- Pharmaceutical sector
- Metal processing sector
- Trade
- Tourism and hospitality
- Construction
- Other:

8. Please indicate up to three critical local factors for entrepreneurship development in the city:

- Special incentive measures provided by the Act on the Renewal and Development of the City of Vukovar
- Positive local entrepreneurial climate
- Proactivity of local authorities
- Quality infrastructure (transportation, digital, energy, etc.)
- Quality city support programs for entrepreneurs
- Quality support programs from the Fund for the Reconstruction and Development of the City of Vukovar for entrepreneurs
- Significant opportunities for using EU funds in the area of the City of Vukovar
- Presence of a Polytechnic with quality educational programs
- Other:

9. Please indicate up to three significant challenges you face in your business that hinder further development:

- Lack of qualified and motivated workforce
- Poor performance of tax and other inspection services
- Low purchasing power of the local population
- Informal economy
- High utility costs (e.g., electricity, gas, waste disposal)
- Inability to collect receivables
- Undercutting by large competing firms
- Poor availability of rental space
- Weak image of the Vukovar economy
- Other:

10. How can the city further enhance the competitiveness of the local economy (rating 1–5, where 1—least important, 5—most necessary):

- Construction and/or municipal development of business zones
- Entrepreneurial incubators and shared workspace facilities
- Upgrading municipal infrastructure for new investments (e.g., water supply and drainage)
- Resolving property-legal issues in locations where there is interest in economic activity
- Stronger promotion of IT education for young people
- Connecting local entrepreneurs into economic-interest groups to strengthen market potential
- Promoting the construction of a logistics distribution centre
- Co-financing the introduction of business mentoring for micro and small businesses
- Increasing the number of entrepreneurial events to strengthen networking among entrepreneurs and promote the city's economy
- Co-financing business rentals in private ownership or reducing rental costs for entrepreneurs in city-owned spaces
- Activating additional business spaces available for rent
- Providing education related to the digital and green transition in the economy
- More active efforts to attract new investors from outside the City of Vukovar
- Installing optical cables throughout the city

- Improving transportation infrastructure
- Developing a sustainable waste management system
- Promoting the construction of renewable energy systems and the use of clean energy in the economy

11. Rating of parts of the call for proposals for the development of entrepreneurship in the City of Vukovar for the year 2021 (rating 1–5, where 1—completely inappropriate, 5—completely appropriate):

- Application conditions
- Volume of documentation to be submitted
- Planned value of the call
- Eligible activities

12. Please provide a final comment on the possibilities for additional stimulation of the economic development of the City of Vukovar by local authorities. ________________

**Appendix B. Interview Questions**

Questions for Entrepreneur Interviews:

1. Briefly describe your company's business, including the category of entrepreneurs you fall into based on size and whether you are an exporter.
2. Can you assess the current business environment for your work? How satisfied are you with the business conditions in Vukovar?
3. What poses the most significant challenge to your business? Can you specify obstacles in your business related to the local level of government?
4. To what extent is transportation infrastructure critical for your business? How do you rate the quality of the transportation infrastructure in Vukovar?
5. To what extent is the quality of the internet important for your business? How do you rate the quality of your internet connectivity?
6. To what extent can you find a suitable workforce in the City of Vukovar? How satisfied are you with their education?
7. In which business segment could the local government help you the most or facilitate your operations?
8. How do you assess the collaboration with the local government? To what extent are you satisfied with the services of city utilities (water supply and sewage, cleanliness, etc.)? Where do you see room for improving the work of the local government?
9. Are you a beneficiary of incentive measures from the City of Vukovar for entrepreneurship? To what extent are you satisfied with these measures? Do you see the need for additional incentive measures by the city authorities, and if so, what measures would those be?
10. Are you a beneficiary of state incentive measures for entrepreneurship (e.g., corporate income tax exemptions)? To what extent are you satisfied with these incentive measures? Do you see the need for additional incentive measures at the state level, and if so, what would they be?
11. How do you assess the perspective of your business in Vukovar, and do you have plans for additional investments and job creation?

Questions for the Croatian Chamber of Economy—Vukovar Branch and the Croatian Chamber of Crafts

1. What are your experiences with business environment and business conditions in Vukovar—what do entrepreneurs mostly complain about and what do they praise?
2. Whish are the most significant obstacles in business, which are related to the local level of government?
3. What are main programmes you offer for local entrepreneurs?
4. How many entrepreneurs are interested in incentives, workshops and educational trainings?

5. What do entrepreneurs say about the quality of transport infrastructure and Internet connection in the area of the City of Vukovar?
6. What is the situation with the labour force in Vukovar?
7. Do you cooperate with the local authorities and in what extent?
8. What are perspectives of entrepreneurship?
9. Other—unstructured, depending on previous answers.

Questions for Fund for the Reconstruction and Development of the City of Vukovar Interview:

1. Can you assess the current business environment in the City of Vukovar?
2. Investments in public infrastructure are visible. In your opinion, why are there not more investments in the Vukovar economy?
3. Can you list the obstacles entrepreneurs complain about the most, especially those related to the local government? Have you had contact with potential foreign investors, and can you assess how they perceived Vukovar as an investment location?
4. How do you assess the state of infrastructure in the City of Vukovar? Is the condition of municipal and transportation infrastructure an advantage or a hindrance to the town of Vukovar?
5. Where do you see the most significant potential for the Vukovar economy? In which sectors?
6. How do you assess young entrepreneurs, or generally, the readiness of young people to engage in entrepreneurship?
7. How do you assess the quality of the local educational system (primary and secondary schools, Polytechnic) in meeting the economy's needs for the workforce?
8. How do you assess the existing incentive measures established by law for entrepreneurs in Vukovar?
9. What new incentive measures for the Vukovar economy would you introduce?
10. How do you assess the projects that entrepreneurs submit to the Fund's calls? To what extent are you satisfied with their quality, and are there qualitative changes over time?
11. How do you assess the cooperation with the local self-government in promoting entrepreneurs?
12. How do you assess the perspective of the economy of the City of Vukovar in the coming years?

Questions for Vukovar Development Agency Interview:

1. Can you describe how VURA promotes entrepreneurship development in the City of Vukovar?
2. How would you rate the quality of project ideas entrepreneurs prepare for public calls (city/national)?
3. What do the most common projects that entrepreneurs apply for public calls look like? Which sectors do they come from, and what are they seeking funding for?
4. How do you assess entrepreneurs' interest in the education that VURA provides?
5. How do you assess the willingness of entrepreneurs to associate and connect (e.g., through clusters, joint project initiatives, etc.)?
6. Can you assess the current business environment in the City of Vukovar? Why, in your opinion, are there fewer investments in the Vukovar economy?
7. Can you list the obstacles entrepreneurs complain about the most, especially those related to the local government?
8. Have you had contact with potential foreign investors, and can you assess how they perceived Vukovar as an investment location?
9. Where do you see the most significant potential for the Vukovar economy? In which sectors?
10. How do you assess young entrepreneurs, or generally, the readiness of young people to engage in entrepreneurship?

11. How do you assess the existing incentive measures for entrepreneurs in Vukovar? What new incentive measures for the Vukovar economy would you introduce (if necessary)?
12. How do you assess the perspective of the economy of the City of Vukovar in the coming years?

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
