# Peer review of "Revitalizing from Ashes: Economic Development and Business Resilience in the City of Vukovar"

_economies, doi:10.3390/economies12020043_

Round 1

Reviewer 1 Report

Comments and Suggestions for Authors

This interesting article deals with post-war economic development of the Croatian city of Vukovar. It focuses specifically on the entrepreneurial sector.

The piece is methodological sound, well written and makes interesting reading, drawing important lessons learned for other towns facing post-war futures. Therefore it is highly recommendable to be published in the journal.

There are very view additional comments to the text, but some issues should be reconsidered by the author(s):

* I found the combination of the discussion and conclusion section unfortunate. I would advice to separate these parts, keeping case-study related info and conclusions to the discussion section and more general observations beyond the case study in the conclusion. There was also no reflection of results in the light of earlier mentioned literature, which would improve the academic merit of the conclusion part considerably.

* Reading the text, I find that the findings for HYP1 + HYP2 are not so unconvincingly put in the discussion and the abstract. Reading the interview and survey parts of the text, I deduct that there are plenty of entrepreneurial incentives, what I see as mainly missing are the 'soft' aspects, such as networking, communication and cross-sectoral exchange. These finding are maybe not highlighted/framed clearly enough in the discussion/conclusion and a partial re-working here could actually make the whole argument stronger.

*There is a fine edge the text walks between the political focus on physical reconstruction and a focus on business support - one thing builds on the other, the text occasionally gives the impression that it sees physical reconstruction as less important. What I found in this context interesting, is that the interviews did apparently yielded some critique on how the rebuilding is taking place - e.g. missing green zones and commercial spaces are reported missing (l. 591 ff). This is found very interesting, as it questions not the reconstruction itself, but maybe more the urban masterplan and its details, an aspect that might deserve a sentence to put it into a better context.

*And finally, there is a text double:  P.4 l.199-209 repeats itself on p.5, l.220-230. 

Otherwise, it was an enjoyable text. Thank you for the contribution.

Reviewer 2 Report

Comments and Suggestions for Authors

The manuscript deals with the important problem of recovering cities from war damages. However, there are some issues that need to be addressed in order to improve the article.

1. Please specify the aim of the research also in the Introduction.

2. Please avoid using expressions like "table/figure below", "table/figure above", "the following table/figure", etc. Use specific table/figure number, wher referencing to it.

3. Please provide the source for Table 1. If it is the result of your work, write "own elaboration".

4. The research sample is quite small. Therefore, we cannot draw meaningful conclusions from it. However, I understand that the city of Vukovar is a quite small one (27k inhabitants), so the large sample would be impossible to obtain. Therefore, please provide the appropriate statement.

5. When describing the research methodology, you describe the Likert scale as the quantitative one. It is not true - Likert scale is the ordinal one.

6. When presenting the results, Figure 1 presents the ration of employed people to total population in Vukovar and in Croatia in total. However, as we look at the employment rate in Croatia in 2021, it says that it is about 68%. On the Figure 1 it is below 41%. I think that you presented the share of employed persons with respect to the total population, which is not correct. Total population also consists of children and retired persons, i.e. economically inactive ones. Therefore, you should present this indicatior as proper employment rate, i.e. share of employed persons (in productive age, which Eurostat presents usually as 20-64 years old) to the total number of economically active population at that age.

7. When you present the structure of selected responses (figures 4-6), you write about percentages. As the sample consisted of just 20 companies, we cannot talk about percentages. percentages should be calculated if the number of units is not smaller than 100. I recommend writing the number of responses, not the percentages.

8. When you present the interview findings and present that some responses received positive rankings - it would be good to include the table with some responses and the median scores. Also, maybe some correlations between the answers would be meaningful.

Round 2

Reviewer 2 Report

Comments and Suggestions for Authors

Dear Authors,

you addressed my comments and remarks to a very satisfactory level. Good job!